# Formulation and Optimization of Repaglinide Nanoparticles Using Microfluidics for Enhanced Bioavailability and Management of Diabetes

**DOI:** 10.3390/biomedicines11041064

**Published:** 2023-04-01

**Authors:** Mubashir Ahmad, Shahzeb Khan, Syed Muhammad Hassan Shah, Muhammad Zahoor, Zahid Hussain, Haya Hussain, Syed Wadood Ali Shah, Riaz Ullah, Amal Alotaibi

**Affiliations:** 1Department of Pharmacy, University of Malakand, Chakdara 18800, Pakistan; saharnasim2003@yahoo.com; 2Center for Pharmaceutical Engineering Science, Faculty of Life Sciences, School of Pharmacy and Medical Sciences, University of Bradford, Bradford BD7 1DP, UK; shahzeb_333@hotmail.com; 3Department of Pharmacy, Sarhad University of Science and Information Technology, Peshawar 18500, Pakistan; hassan.fls@suit.edu.pk; 4Department of Biochemistry, University of Malakand, Chakdara 18800, Pakistan; mohammadzahoorus@yahoo.com; 5Department of Pharmaceutics and Pharmaceutical Technology, College of Pharmacy, University of Sharjah, Sharjah 27272, United Arab Emirates; zhussain@sharjah.ac.ae; 6Research Institute for Medical and Health Sciences, University of Sharjah, Sharjah 27272, United Arab Emirates; 7Department of Pharmacy, Shaheed Benazir Bhutto University, Sheringal Dir 18000, Pakistan; haya@sbbu.edu.pk; 8Medicinal Aromatic and Poisonous Plants Research Center, Department of Pharmacognosy, College of Pharmacy, King Saud University, Riyadh 11451, Saudi Arabia; rullah@ksu.edu.sa; 9Department of Basic Science, College of Medicine, Princess Nourah bint Abdulrahman University, Riyadh 11671, Saudi Arabia; amaalotaibi@pnu.edu.sa

**Keywords:** repaglinide (Rp), microfluidic technology, in-vitro, in-vivo study, bioavailability, rats

## Abstract

The technologies for fabrication of nanocrystals have an immense potential to improve solubility of a variety of the poor water-soluble drugs with subsequent enhanced bioavailability. Repaglinide (Rp) is an antihyperglycemic drug having low bioavailability due to its extensive first-pass metabolism. Microfluidics is a cutting-edge technique that provides a new approach for producing nanoparticles (NPs) with controlled properties for a variety of applications. The current study’s goal was to engineer repaglinide smart nanoparticles (Rp-Nc) utilizing microfluidic technology (Dolomite Y shape), and then to perform in-vitro, in-vivo, and toxicity evaluations of them. This method effectively generated nanocrystals with average particle sizes of 71.31 ± 11 nm and a polydispersity index (PDI) of 0.072 ± 12. The fabricated Rp’s crystallinity was verified by Differential scanning calorimetry (DSC) and Powder X-ray diffraction (PXRD). In comparison to the raw and commercially available tablets, the fabricated Rp’s nanoparticles resulted in a higher saturation solubility and dissolving rate (*p* < 0.05). Rp nanocrystals had a considerably lower (*p* < 0.05) IC_50_ value than that of the raw drug and commercial tablets. Moreover, Rp nanocrystals at the 0.5 and 1 mg/kg demonstrated a significant decrease in blood glucose level (mg/dL, *p* < 0.001, *n* = 8) compared to its counterparts. Rp nanocrystals at the 0.5 mg/kg demonstrated a significant decrease (*p* < 0.001, *n* = 8) in blood glucose compared to its counterparts at a dose of 1 mg/kg. The selected animal model’s histological analyses and the effect of Rp nanocrystals on several internal organs were determined to be equivalent to those of the control animal group. The findings of the present study indicated that nanocrystals of Rp with improved anti-diabetic properties and safety profiles can be successfully produced using controlled microfluidic technology, an innovative drug delivery system (DDS) approach.

## 1. Introduction

Persistent hyperglycemia is a sign of diabetes mellitus (DM), a chronic endocrinological disorder [1]. In the past 35 years, the prevalence of DM has grown by nearly four times [2]. The International Diabetes Federation’s most recent statistics report indicates that there are currently more than 500 million individuals worldwide who have diabetes. It is predicted that this figure will rise to 643 million by 2030, or 11.3% of the world’s population, and to 783 million by 2045 [3,4]. The accompanying life-threatening consequences, including cardiovascular, cerebrovascular, and renal problems, made DM the ninth most common cause of death in 2019 [5]. Due to the sharp increase in the prevalence of DM, scientists from both academia and the R&D sectors have been focusing on the development of new anti-diabetic drugs and modifying physicochemical attributes of the existing therapeutics to effectively treat and control this challenging disease [6,7].

The Biopharmaceutical Classification System (BCS) class II drugs glipizide (GPZ), gliclazide (GCZ), tolbutamide (TOL), Repaglinide (Rp) and glibenclamide (GBC) for diabetes have low water solubility. Repaglinide (Rp) is an oral antihyperglycemic medication of the meglitinide family that is used to control postprandial hyperglycemia in the treatment of type II diabetes. In the therapeutic management of patients with a novel chemical family of insulin secretagogues, Rp is a highly favored therapeutic alternative to sulfonylureas [8]. Owing to strong and variable hepatic first-pass impact, the pharmacokinetic study states that Rp has a short biological half-life (1 h) and limited bioavailability (56%) [9]. However, repaglinide treatment has not been shown to be associated with toxicological aspects in preclinical and clinical studies [10]. Many people using this medication do not have serious side effects, but some mild effects like gastrointestinal disturbance, headache, and musculoskeletal pain have been reported [11]. The safety profile of Repaglinide in animal models has been established and did not have significant oral toxicity with an LD_50_ > 3000 mg/kg [12]. The effective dose for preclinical and clinical studies in animal models and humans has been reported to Repaglinide as 0.5, 1 and 2 mg [13,14].

The oral route of drug administration has been reported to be one of the safest routes for a range of drug compounds. However, poor water solubility and bioavailability have been reported the hard barriers for >90 new drug compounds coming from synthesis and high throughput screening [15]. Nanotechnology is the field of science that deals with the synthesis and exploitation of nanomaterials for different applications. The physicochemical properties of the active pharmaceutical ingredients are changes on the nanoscale, which in turn results in significant improvement in their bioavailability and physiological activities [4].

According to the Biopharmaceutical classification system and on the basis of solubility and intestinal permeability of the pharmacological substances, pharmaceutical ingredients are classified into four classes as BCS-I-BCS-II, BCS-III, and BCS-IV. Amidon and his colleagues developed this technique in 1995 to eliminate the necessity for in-vivo bioequivalency tests [16,17]. Low solubility and high permeability are known as molecular properties of BCS class II agents [18]. The bioavailability of BCS class II drug compounds is limited by dissolution rate, and a minor increase in dissolution rate of BCS-II drug substances can significantly increase their potential bioavailability [19]. The bioavailability, which can be increased by a decrease in a drug’s particle size, is defined as the percentage of the drug’s amount that is absorbed relative to its initial dosage [20]. In particle engineering and drug kinetics prospectives, the dissolution rate of the BCS-II drug substance is inversely proportional to particle size; thus, a small particle size, especially at the nano size level can significantly enhance bioavailability of the said drug compounds [21]. In order to suit the demands of biopharmaceutical and processing, the drug delivery system generally requires narrow particle size distribution with consistent particle shape [22]. Fullerenes, quantum dots, nanotubes, nanopores, liposomes, dendrimers, radio-controlled nanoparticles, nanocrystals, microfluidics, and magnetic nano probes are some of the well-established nanotechnological platforms being explored for different applications [23]. Amongst these approaches, microfluidics have got a noticeable attraction for novel drug delivery systems.

The production of nanoparticles (NPs) using conventional approaches is susceptible to batch-to-batch fluctuations and polydispersity of the engineered particles. However, in Microfluidics technology, where flow of the feeding solutions through nano/picoliter size channels is controlled, which in turn produces the smart nanoparticles. During the past 10 years, microfluidics-based technologies have become an effective and advanced bottom up technology for engineering of a range of challenging drug compounds to treat many challenging diseases [24,25].

The current study aimed to exploit Dolomite Y shape microfluidics technology through a controlled process and under experimental conditions for engineering of smart nanoparticle of repaglinide with enhanced antidiabetic potential. Particle size, scanning electron microscope (SEM), Differential scanning calorimetry, X-ray diffraction, Transmission electron microscopes (TEM), in-vitro dissolution, and an in-vivo antidiabetic stability investigation in a rat model were used to characterize the repaglinide nanocrystals.

## 2. Materials and Methods

### 2.1. Materials

Repaglinide (Rp) was kindly gifted for research purposes by Wilshire Laboratories-Pvt Ltd., Lahore, Pakistan, with the Batch No. (RG 1912130216) supplier, Aurobindo Pharma Ltd., Hyderabad, India. PVP-K30 (Polyvinylpyrrolidon, 08297052G0), HPMC (hydroxypropylmethyl cellulose, 8028213), PVA (polyvinyl alcohol), Pluronic 127 (P-127), and Pluronic F68 (P-F68) were purchased from BASF, Ludwigshafen, Germany. Tween-80 (9005-65-6), Streptozotocin (STZ, 18883-66-4), and SLS (sodium dodecyl sulphate, 08421LE) were procured from Sigma Aldrich, Gillingham, UK.

### 2.2. Preparation of Repaglinide Nanosuspension

Repaglinide (Rp) nanocrystals were prepared by using Dolomite^®^ microfluidics reactor (MFR) Royston, Hertfordshire, UK (100 µm). Briefly, Rp 25 mg/mL was dissolved in ethanol. Solvent containing Rp was introduced through syringe pump into aqueous solution comprised of different stabilizers/surfactants, including PVP-K30, HPMC, PVA, Pluronic 127, P-F68, and Tween 80 in different concentrations (0.5%, 1%, 1.5% and 2%). For optimization of the process and experimental conditions, which include flow rate of solvent antisolvent, stabilizer concentration, types of stabilizer, various nanosuspensions formulations were produced at a milliliter scale (Table 1 and Table 2). The mean particle size and distribution of each nanosuspension formulation was measured using a dynamic light scattering instrument (DLS) Malvern Zetasizer Nanoseries (Malvern Instrument, Malvern, UK).

### 2.3. Characterisation, Stability, and Dissolution Studies of Repaglinide Nanocrystals (Rp-Nc)

Repaglinide (Rp) nanocrystals were subjected to different characterisations, including particle size measurements, melting point detection, crystallinity, particles morphology stability, and dissolution studies.

#### 2.3.1. Determination of Particle Size

The particle sizes at mean value of nanocrystals (Rp-Nc) were ascertained by Malvern, Zeta-sizer, Nano-.ZS, UK. The sample for particle size measurements were prepared according to the method reported by Plakkot et al., (2011). One milliliter of the sample was further diluted with water (1:1). All the samples were analyzed in triplicate (*n* = 3) [26].

#### 2.3.2. Determination of Zeta Potential

Zetasizer Nano-ZS was used to determine the zeta potential of repaglinide nanosuspension. The samples for assessment were produced as per reported protocol [26]. The developed nanosuspensions samples were diluted further, using the dispersion medium. The original samples of the produced nanosuspension were further diluted with water (1:1) and analysed by Malvern, Zeta-sizer, Nano-.ZS, UK. All the samples were analyzed in triplicate and measured mean ± S.D.

#### 2.3.3. Scanning Electron Microscopy

For the morphological studies, the Rp micronized drug, SEM (Quanta 400 series, Cambridge, UK) was used. Photomicrographs of RG were taken at different ranges of magnifications. Before processing, the samples were coated with gold.

#### 2.3.4. Transmission Electron Microscopy

Particle morphology of the produced Rp nanoparticles was assessed using the transmission electron microscope (TEM) (TEM, Ex200, Tokyo, Japan) at 120 KV. The RG deposited sample grid (coated formvar/carbon grid with cooper, having mesh size 200) was dried at ambient temperature. Two percent magnesium uranyl acetate was used for negative staining of the grid on which the sample was deposited.

#### 2.3.5. Differential Scanning Calorimetry

Determination of the melting point and the heat of fusion of unprocessed Rp and its fabricated nanocrystals was carried out by Differential scanning calorimetry (DSC) (Q2000, TA Instruments, Crawley, UK). The unprocessed Rp and fabricated samples were processed using the nitrogen gas at a flow rate of 50 mL/min, and the temperature range was adjusted to 25–200 °C with a heating rate of 5 °C.

#### 2.3.6. Powder X-ray Diffraction Studies

Crystallinity of the processed and raw Rp samples was analyzed by *P-XRD* (Bruker, Germany). All the samples were loaded in silicon wells followed by scanning over the range of 5–500° at 2Ɵ at the rate of 10° 2Ɵ)/min and wavelength (1.542 Ǻ).

#### 2.3.7. Saturation Solubility Studies

Saturation solubility attributes of the unprocessed Rp and Rp nanoparticles were determined and compared both in pure water and in a stabilizer solution containing PVA (1%) and P-F127 (2%) *w*/*w*. The samples were centrifuged at 1600 Rpm for one hour [27,28,29]. To ensure the complete solubility of the developed nanocrystals in all the samples, the supernatant layer from the developed sample was filtered using Syringe Filter (0.02 µm: 20 nm). The filtrates were assayed for the RP with HPLC by using the Acetonitrile, Solution A, and methanol Mobile phase in (49:40:11), respectively, at excitation wavelength 244 nm using a fluorometric detector at flow rate of 1 mL/min and injection volume 20 µL using Column: 4.0-mm × 12.5-cm; 10-µm packing L1 (C18).

#### 2.3.8. Assessment of Stability Studies

The developed nanosuspensions of Rp were subjected to physical stability by determining the particle size and zeta potential that are stored at various temperatures (2–8 °C, 25 °C and 40 °C) for 3 months using Malvern, Zeta-sizer, Nano-.ZS, UK.

#### 2.3.9. Dissolution Studies

The dissolution profile of the developed Rp nanocrystals (Rp-Nc) was compared with unprocessed Rp powder, marketed tablets of Rp and microsuspension (11.5.0 µm ± 3.25 micron). The microsuspension was prepared from the marketed tablets by crushing with pestle and mortar followed by suspending in aqueous stabilizer medium. The unprocessed API with the mean particle size 50 µm ± 2.33 was used for comparison. The medium (9000 mL) for dissolution study consisted of buffer pH 5.0. The medium was prepared by dissolving 10.25 gm of citric acid monohydrate salt and dibasic sodium phosphate 18.16 gm in 1000 mL of purified water using the reported method as described in USP 366-NF31. The speed of the dissolution apparatus was set at 75 rpm utilizing the paddle method (USP apparatus II). For estimation of the drug contents, 5 mL of aliquots from the dissolution medium were collected at various time intervals at temperature 37 ± 0.5 °C using 0.02 µm size filter disc. To retain the sink conditions, 5 mL fresh dissolution medium was added after each aliquots sample [30,31]. Drug contents for the Rp was analyzed using HPLC at the flow rate of 1.0 mL/L using the method as reported in USP 366-NF31 at excitation wavelength 244 nm and excitation wavelength of 348 nm using Fluorometric detector.

### 2.4. Animals

Eight to ten-week-old male Wister rats weighing 170–200 g and Balb/C mice weighing 20–25 g were procured from VRI (Veterinary research institute), Lahore and were placed in the animal house under standard conditions for laboratory (temperature = 25 ± 2 °C, relative humidity = 55–65% and light/dark cycle = 12 h) water and standard diet was also provided. Prior to the experiments, animals were allowed to be adapting to laboratory conditions and were dealt with as per procedures stated in the “Animals Byelaws 2008 of University of Malakand (Scientific Procedures Issue-I)”. The study was carried out as per approval of Ethical Committee of the Department of Pharmacy, vide notification no: Pharm/EC-Rpg-Nc/44-09/22.

### 2.5. Assessment of Acute Toxicity Study

The acute toxicity of Rp-Nc was carried out in two phases according to the Lorke method with slight modification. Oral doses of Rp-Nc in mg/kg body weight and Tween-80 were given to control groups in the first phase and 2nd phase. In the first phase, one group (control group) was given Tween-80 (2%) and various oral doses (10, 75 and 150 mg/kg body weight were given to the remaining groups, respectively. In the first phase, various oral doses (250, 500 and 1000 mg/kg body weight were given to the remaining groups, respectively. Animals were kept under examination for 24 h initially and then on a daily basis to observe signs of convulsions, diarrhea, sleeping, lethargy, tremor, and salivation. Mortality of animals were also monitored if it occurred [32]. Effective doses for pharmacological activities (0.5 and 1 mg/kg b.w) were selected as per preliminary pharmacological studied as well as published data elsewhere [13,14].

### 2.6. Induction of Type II Diabetes

After adaptation, 50 mg/kg of streptozotocin (STZ) in 0.1 M citrate buffer was injected intraperitoneally (i.p.) to the overnight fasted HFD rats. In order to avoid deaths by hypoglycemic shock, glucose solution (10%) was given to animals for about three days. Post 72 h of STZ administration, blood glucose level was checked by taking blood from the tail vein and glucose level was determined by using SD glucometer (ACCU-CHECK, Active blood glucose meter, Korea). Those animals having fasting blood glucose level higher than 250 mg/dL were considered to be diabetic [33].

### 2.7. Experimental Design for Assessment of Antidiabetic Activity

The total of 48 animals after induction were categorized into six groups, each containing eight animals (*n* = 8), comprising diabetic, control, test groups treated with 0.5 and 1 mg/kg b.w of Rp-Nc; unprocessed Rp (API) and marked tablets (1 mg/kg) were administered. The vehicle was administered to animals in STZ diabetic groups and in control (normal) group. The treated animals received Rp-Nc p.o. at their respective doses (0.5 and 1 mg/kg) for four weeks.

### 2.8. Estimation of Blood Glucose Level and Body Weight

Blood glucose level and body weight were taken on the first day of each of the four weeks of treatment [34].

### 2.9. Assessment of Serum Profile

Upon completion of the antidiabetic assay on the 28th day of the study, all of the animals were sacrificed with isoflurane in a humane manner and blood samples were obtained by cardiac puncture for assessment of the biochemical parameters like serum alkaline phosphatase (ALP), High density lipoprotein (HDL), Low density lipoproteins (LDL), Total cholesterol (TC), Triglycerides (TGs), insulin level and HbA1c%, which were determined. The oxidative stress marker and antioxidant enzyme like Superoxide dismutase (SOD), catalase (CAT), and lipid peroxidation (MDA) were also determined. The vital organs including liver, kidney, and pancreas of animals used in groups were embedded in paraffin, sectioned with microtome followed by staining with hematoxylin and eosin stain (H&E), and examined under a light microscope (×40) [35].

### 2.10. Statistical Analysis

All data were expressed as mean ± SEM (*n* = 8). The significance of variations were estimated using Graph Pad Prism version 5.01 (San Diego, CA, USA), using one way ANOVA followed by Dunnett’s post hoc multiple test.

## 3. Results and Discussion

### 3.1. Fabrication, Optimisation and Characterisation of Rp Nanoformulation

The approaches employed for development of repaglinide (Rp) nanosuspension were with the use of a Dolomite microfluidic chip. Wherein, the Rp 25 mg/mL was dissolved in ethanol and the solvent and antisolvent were injected using a syringe pump at different flow rates. Different stabilizers were used in order to find the most appropriate stabilizer and flow rate (Table 1). The least particle size (P.S) 169.2± and polydispersity (PDI) 0.32 ± 34 was resulted initially at the flow rate of 50 µL/min and 80 µL/min by using PVA 1% *w*/*w*, which was further optimised (Table 2). For the optimized Rp nanoparticles, the most effective stabilizer was found to be PVA (1% *w*/*w*) resulting in the mean P.S. of 71.31 ± 11 nm and having a narrow PDI (0.072 ± 12) by using Malvern, Zeta-sizer, Nano-.ZS, UK, whereas the flow rate of solvent and antisolvent was 50 µL/min and 120 µL/min, respectively (Table 1). Furthermore, by increasing the flow rate, both the PDI and the PVA were increased (Table 2). A marked size reduction of Rp was achieved from the initial size of the unprocessed repaglinide 50 µm, which is shown in (Figure 1a). TEM images of Rp were captured at a magnification power of 80 K, and the average particle size was found to be 65 nm (Figure 1b). As illustrated in (Figure 1b), which confirms the particle size of Rp being on the nanometre scale, consistent with the dynamic light scattering (DLS) determination.

In microreactors, drug in the solution form and antisolvent (water) stream flows parallel devoid of turbulence (laminar fashion) with the development of a central diffusion layer at the interface between them [36]. The process of nanoprecipitation in microreactors is commenced by a combination of the Rp solution streams and the antisolvent in such a manner that supersaturation of the drug, nucleation, and the particle growth take place in a central diffusion layer [37].

The flow rate of the antisolvent (water) could be tuned to direct the particle size of the developed product with antisolvent flow at higher rate values than that of the drug solution rate to achieve maximum supersaturation and induction of drug nucleation ensuing in particles size reduction. Additionally, decrease in particle growth is observed by increasing the flow rate of antisolvent due to minimizing the solute availability around the drug particles for growing [38].

For crystal size growth and distribution, the optimum selection and concentration (proportion) of the growth stabilizer/inhibitor is a crucial parameter. Adsorbed stabilizers prevented the crystals from growing, and for adequate coverage, the stabilizer concentration should be sufficient to completely cover the crystals’ surfaces and produce a strong enough steric force to repel them. Agglomeration and crystals growth can occur due to inadequate coverage of the crystals by stabilizer, while high concentration can hinder transmission of ultrasonic vibration, due to high viscosity that adversely influence the diffusion between solvent and anti-solvent [39].

### 3.2. Stability Studies

The stability of optimum formulations of Rp nanocrystals (Rp-Nc) were monitored at different storage temperatures (05, 25 and 40 °C) for one month at (0, 10, 15, 20, 25 and 30th days). As shown in the (Figure 2a), Rp nanosuspension was found to be stable in refrigerated condition (5 °C) and at room temperature (25 °C) (Figure 2b) with slight increase in the size at end of day 15, having no significant change in the particle size and distribution over one month, confirming the stable nature of the fabricated nanocrystals by microfluidics. Although, at 40 °C (Figure 2c) increased in particle size and its distribution was observed, which was most likely due to the Ostwald ripening at a high temperature, causing the increase in particle size [40]. The level of supersaturation and fast nucleation is anticipated to increase due to the probable decrease in kinetic energy and diffusion at low temperatures— given that the number of nuclei rose as the quantity of solute on each nucleus dropped [38].

Zeta potential measurement of formulated nanosuspensions has been reported to be paramount for physical stability. The measurements of Zeta potential rests on both surface of the particles and the composition of stabilizer used to a medium extent. A range of minimum zeta potential values for electrostatic and steric stabilization of nanosuspensions has been reported, which include −30 mV and −21 mV, respectively [41,42]. For fabrication of stable nanosuspension, both ionic and non ionic polymers and surfactants as stabilizer with minimum zeta potential value -21 has been proposed [37]. These values of zeta potential of Rp nanosuspensions were observed within the reported range (±21.0), and no significant difference was observed in different formulations. The resulting zeta potential also demonstrated that PVA has effectively adsorbed onto the surface of Rp Nanocrystals (Rp-Nc). Physical stability studies of Rp nanocrystals at refrigerated condition, room temperature, and elevated temperature i.e., 5 °C, 25 °C and 40 °C, respectively, for 30 days showed that nanocrystals stored in a refrigerated condition (5 °C) and at room temperature (25 °C) were extremely stable compared to the samples stored at 40 °C (Figure 3). The fabricated nanocrystals stored at refrigerated conditions for one month displayed the greatest stability by keeping PDI values with homogeneous particle size distribution at 5 °C, 25 °C *p* ≤ 0.05, with subsequent avoidance of Ostwald ripening in produced nanosuspensions [40].

Temperature has been documented to have a key role in physical stability of suspensions, because kinetic energy of suspended particles increments with high temperature that results in strong van-der Waals forces among the particles with consequent agglomeration and de-stabilization of the suspensions [43]. For maximum stability of nanosuspensions Frietz and Muller 1998 suggested a storage temperature of 2–8 °C [44]. In addition, rapid particle growth occurs with increase in temperature and high intensity of light radiation. Owing to high solubility of the drug compounds at high temperature, the supersaturation ratio which is shown in equation 2 decreases, which leads to a decrease in nucleation rate followed by rapid particle growth.

### 3.3. DSC and P-XRD Studies

For determination of thermal profile and crystallinity Differential scanning calorimeter (DSC) and Powder x ray diffraction (PXRD) have always been the most promising techniques in term of assessing the solid samples.

DSC and PXRD studies of the pulverized Rp particles demonstrated that Rp retained its physical form and crystallinity. For both nanocrystals and unprocessed Rp, a single sharp melting endotherm was noted (Figure 3). However, the melting point peak of the raw RP appeared at a slightly higher temperature (130.5 °C) compared to processed sample (124 °C). Secondly, the endothermic peak of the fabricated sample was slightly broadened. These differentiations can cause the differences between the particle sizes of both raw and produced nanocrystals, because DSC profile can be unequivocally influenced by the size of the particles and packing density and nanoparticles compared to raw sample that has a low enthalpy and melting point [45,46]. Additionally. broadening of the DSC peaks may be due to impurities or to traces of the polymers/surfactants that remained on the surface of the drug particles [46].

The PXRD investigation of both unprocessed Rp and nanocrystals produced sharp X-ray chromatograms which validate that Rp maintained its crystalline nature (Figure 4). However, a broad PXRD peak was observed, with the disappearance of most of the peaks, which reflects the semi-crystalline nature of the produced Rp nanoparticles. If there appeared a straight line in the XRD analysis, it would demonstrate the pure amorphous nature of the engineered particles. For nanocrystals X-ray diffractograms with less peak intensities and disappearance of some peaks have been previously reported [30,47,48]. Broadening and vanishing of some peaks may appear for smaller particles due to small angle reflection by the particles that shift the peak intensity to a lower level [45].

### 3.4. Solubility Studies

Solubility studies of Rp nanocrystals (Rp-Nc) and unprocessed Rp in pure water and stabilizer solution are given in Figure 5. For the Rp nanocrystals the solubility was reported to be 86.1 µg/mL ± 1.33, an almost 2.3-fold and 4.1-fold increase compared to the solubility of unprocessed Rp and Rp in stabilizer solution (37.3 µg/mL ± 2.3) or in pure water (20.8 µg/mL ± 2.0). Due to the enlarged surface area of the fabricated nanocrystals, a significant change (*p* ≤ 0.05) in solubility of the fabricated Rp was observed compared to the solubility in the water and stabilizer solution. Additionally, a rise in the solubility of Rp in stabilizer solution using PVA has also been observed.

However, the nanocrystals exhibited a significant increase in saturated Rp solubility showing that the Rp solubility was increased only due to the small particle size of the particles and not by the polymer or surfactant. In general, improving the saturation solubility of a drug can be achieved by reducing the particle size of the drug or changing its crystalline state, for instance, forming amorphous particles.

### 3.5. Dissolution Studies

For in-vitro studies, the dissolution rate of the repaglinide unprocessed and micronized was compared with repaglinide nanosuspension. As illustrated in (Figure 6), a significant change in the dissolution rate of Rp nanosuspension was documented and was found to be considerably faster compared to its unprocessed Rp and micronized drug. The release pattern of the nanosuspension showed a significantly faster release rate of the drug. The raw repaglinide showed very poor diffusion with nearly 22.5% of drug release in 30 min, while its nanosuspension showed a faster release rate, with 67% of the drug defused in 2 min and 100% in 30 min. For nanosuspension, statistics showed significant (*p* < 0.05) change in the in-vitro studies compared to unprocessed (raw) and micronized drugs. This increased rate of dissolution is reassuring, due to the smaller particle size present, the large surface area with a high rate of dissolution ensuing, and the faster absorption rate, eventually resulting in the maximum bioavailability of the drug. It has been documented that with an increase in saturation solubility, more specifically at a range of 100 nm particle size, the dissolution rate happens to be faster than described [49].

### 3.6. Effect on Blood Glucose in Induced Diabetes

The animals in the STZ group showed significant elevation in blood glucose levels compared to the normal group (*p* < 0.001, *n* = 8). For the period of treatment of 28 days, the level blood glucose in all groups of rats was checked at time intervals (every week). Due to STZ administration, the untreated (diabetic) group had considerably raised blood glucose levels compared to the normal group of animals. During the first week of administration, no significant decline in the glucose level of groups was noticed. Comparatively, the samples treated group after the second and third weeks (14 and 21 day) of treatment had reduced blood glucose to a significant degree; these levels were found to b 276.90 ± 4.67 mg/dL and 239.75 ± 4.82 mg/dL (*p* < 0.001, *n* = 8) for 0.5 and 1 mg/kg, respectively, on the 14th day in comparison to the diabetic control group (504.39 ± 4.91 mg/dL, *p* < 0.001, *n* = 8). While a significant decline in the blood glucose level was observed on day 21st and was found to be 170.81 ± 4.71 mg/dL and 142.49 ± 4.66 mg/dL (*p* < 0.001, *n* = 8) for 0.5 and 1 mg/kg, respectively, in comparison to the diabetic control group (490.72 ± 5.02 mg/dL, *p* < 0.001, *n* = 8). At the end (28 days), the glucose levels of treated groups compared to the diabetic group, there was a significant decline observed in their glucose level, which were almost near the values of normal control group (Table 3).

The effect of exposure to treated samples on body weight of rats was studied over the period of 28 days. In the diabetic group, the body weight increased significantly as the diabetic group did not receive any treatment, while there was no significant change in the body weight of the treated groups in comparison to control group (Table 4).

### 3.7. Antihyperlipidemic Effects of Rp-Nc on Induced Diabetes

In this study, STZ administration resulted in the elevation of plasma total cholesterol, LDL, triglycerides and lowered the level of HDL in comparison to the normal group (Table 5).

Samples treatment significantly revert the lipid profile level. Plasma total cholesterol [F (4, 35) = 453.2, *p* ˂ 0.001, *n* = 8)], LDL and triglycerides levels [F (4, 35) = 236.0, *p* ˂ 0.001, *n* = 8)] in diabetic groups treated with samples were reduced significantly (Figure 7). While HDL level was considerably increased in all treated groups when compared to the diabetic group (Table 5). Consequently, the samples are anticipated to have hypolipidemic activity.

As shown in Table 5, it was understandable that STZ induced diabetic group produced a marked decline in the insulin level. Sample-treated diabetic groups showed significantly higher insulin level [F (4, 35) = 87.49, *p* ˂ 0.001, *n* = 8)] compared to normal control group.

In the diabetic control group of animals, a significant elevation in the HbA1c level (*p* < 0.001, *n* = 8) was observed. The sample-treated diabetic groups had significantly reduced blood levels of HbA1c in comparison to control group (Table 6).

### 3.8. Effects on Liver

The abnormality in the liver induced by STZ has been ascribed to variation in serum alkaline phosphatase (ALP) and serum glutamate pyruvate transaminase (SGPT). Test samples were screened for liver function tests that caused a decrease in the serum ALP and SGPT indicating hepatic tissue damage reversal and plasma membrane stabilization. The findings revealed that the samples administration significantly (*p* < 0.05, 0.01 and 0.001, *n* = 8) reduced the serum ALP and SGPT elevated levels (Table 7).

In STZ induced diabetic group, the MDA level was reported to be higher as an end product of lipid peroxidation due to excessive production of free radicals that would lead to tissue degeneration. Sample-treated diabetic groups significantly decreased the MDA level considerably as shown in Table 8.

While in sample treated groups the activity of SODs were increased significantly to the normal level [F (4, 35) = 37.06, *p* ˂ 0.001, *n* = 8)] (Table 8 and Figure 8) and similar type of findings were observed in the catalase (CAT) activity, revealing the protective effect that could be associated with antioxidant properties.

Photomicrograph of kidney sections stained with hematoxylin and eosin stain (H&E) from the histopathological investigation examined under light microscope (×40) of normal control group (Figure 9) revealed normal nephrons, renal capsule and tubules in comparison to diabetic control group showing mild to moderate degeneration in renal tubular portion in association with mononuclear cells infiltration resulting in epithelial desquamation and necrosis.

The photomicrograph of liver histopathology from normal control group revealed normal architecture in comparison to mild hydropic degeneration of hepatocytes in diabetic control group with mild necrosis and infiltration (Figure 10). Administration of Rp-Nc at dose concentrations showed some improvement in the architecture of renal capsule, nephrons and tubules and improvement in hepatic architecture towards normal.

The photomicrograph of pancreas of rat histopathology from normal control group revealed showing the normal histological structure of islets of Langerhans with normal pancreatic acini in comparison to ruptured and destructed islets of Langerhans with damage in beta cells in diabetic control group (Figure 11). Administration of Rp-Nc at dose concentrations showing some normal islets of Langerhans with some mild destructed in normal pancreatic duct and in between normal pancreatic acini towards normal.

## 4. Conclusions

Repaglinide (Rp) nanocrystals having mean particle sizes of 71.3 ± 1.3 nm and polydispersity index of 0.072 were successfully fabricated by using microfluidic to increase the drug’s oral bioavailability. A high stability was achieved for Repaglinide (Rp) nanocrystals when stored at 5 °C and 25 °C for three months. Both in-vitro and in-vivo characterizations of the produced Rp nanoparticles were performed and compared with the raw counterparts. The blood glucose level, body weights, glutamate oxaloacetate transaminase (SGOT), Low density lipoproteins (LDL), High density lipoprotein (HDL), glutamate pyruvate transaminase (SGPT), lipid peroxidase, Triglycerides (TGs), Total cholesterol (TC), serum insulin, and HbA1c levels were measured in both normal and diabetic rats. According to in-vivo findings, therapy with Rp-Nc corrected serum biochemical parameters and blood glucose levels in comparison to streptozotocin induced diabetic control group. The protective effect was further supported by histological analysis. Following treatment with Rp-Nc, these rats had a similar histological regeneration. The results were significant, showing the efficacy of microfluidic technique for the preparation of Rp-Nc by increasing Rp oral bioavailability and supported its potential for the management of diabetes.

## Figures and Tables

**Figure 1 biomedicines-11-01064-f001:**
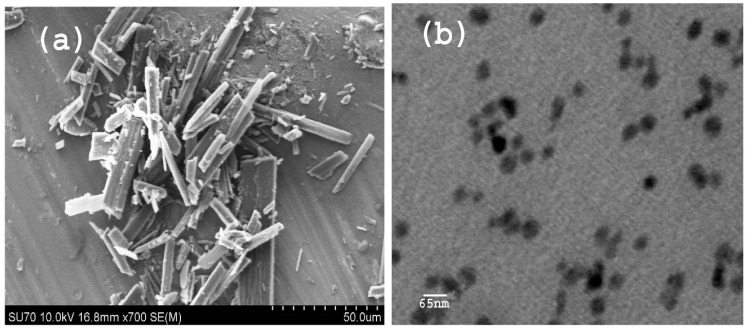
Images of Rp (**a**) and (**b**) at different magnification.

**Figure 2 biomedicines-11-01064-f002:**
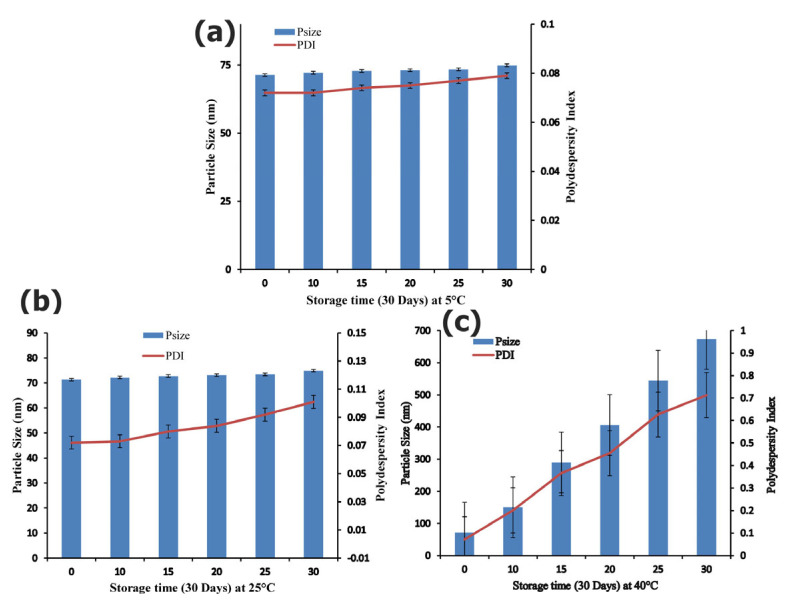
Stability study at different storage temperature; (**a**) at 5 °C (**b**) at 25 °C and (**c**) at 40 °C for 30 days.

**Figure 3 biomedicines-11-01064-f003:**
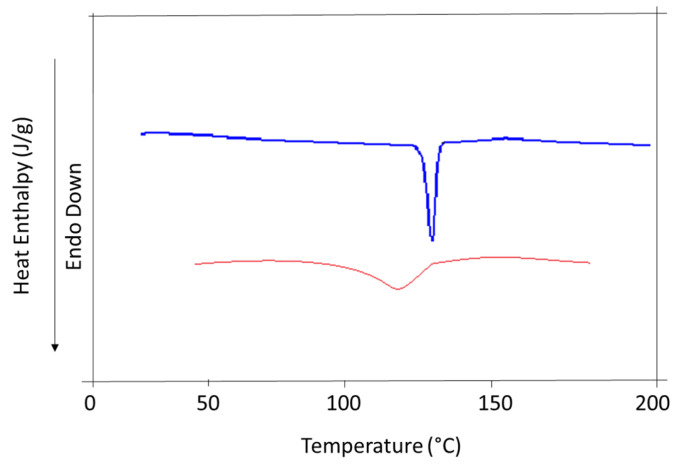
DSC thermograms of unprocessed (raw) and nanocrystals.

**Figure 4 biomedicines-11-01064-f004:**
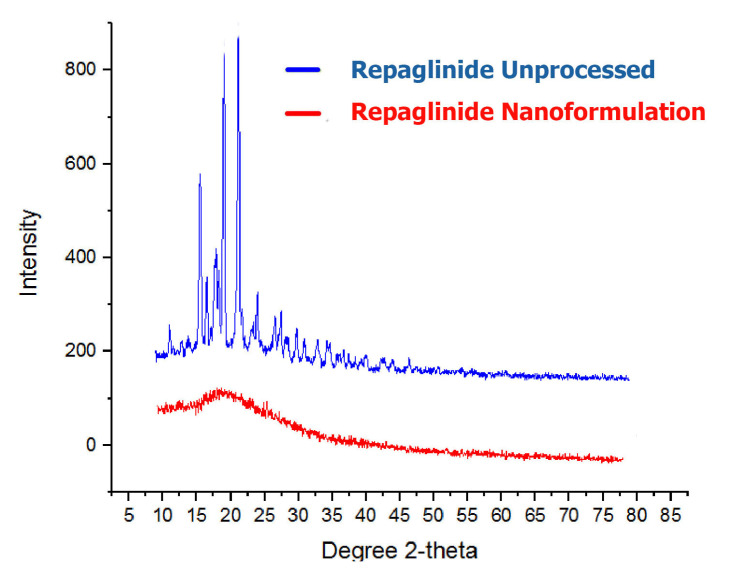
PXRD peaks of unprocessed Rp and Rp-Nc.

**Figure 5 biomedicines-11-01064-f005:**
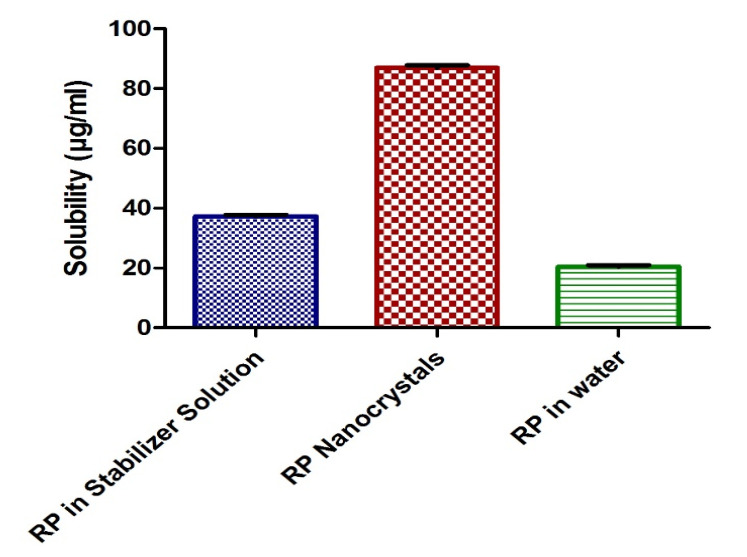
Solubility studies of Rp nanocrystals (Rp-Nc) and raw Rp.

**Figure 6 biomedicines-11-01064-f006:**
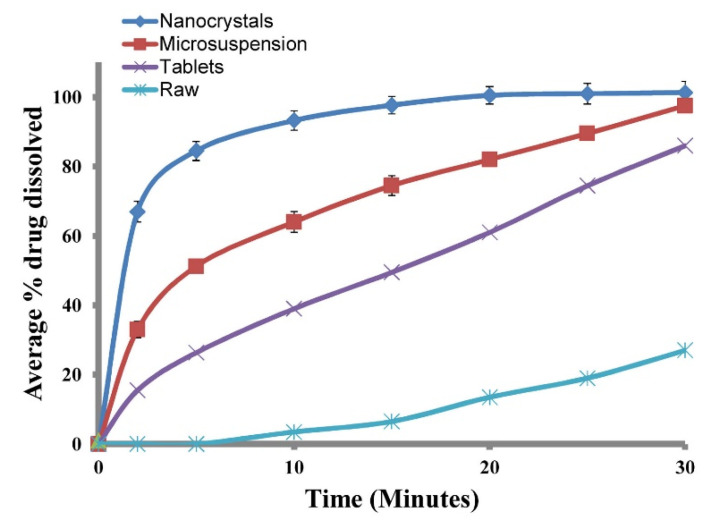
In-vitro dissolution studies of the repaglinide.

**Figure 7 biomedicines-11-01064-f007:**
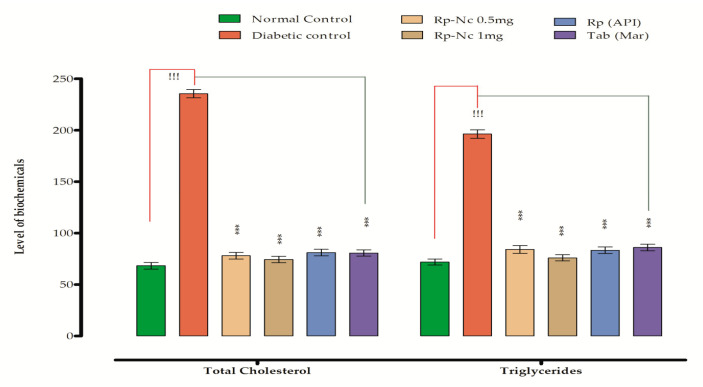
Effect of Rp-Nc on level of biomarkers. Mean ± SEM (*n* = 8). Significance was determined statistically using one way ANOVA following Dunnett’s comparison test,^!!!^ *p* < 0.001 vs. normal control group and (*** *p* < 0.001, *n* = 8) by vs. diabetic control group.

**Figure 8 biomedicines-11-01064-f008:**
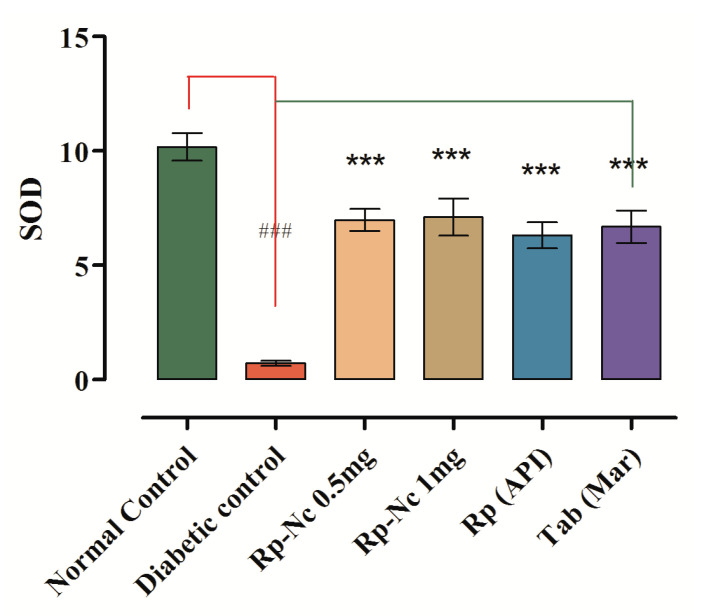
Effect of Rp-Nc on SOD level. Mean ± SEM (*n* = 8). Significance was determined statistically using one way ANOVA following Dunnett’s comparison test, ^###^ *p* < 0.001 vs. normal control group and (*** *p*< 0.001, *n* = 8) by vs. diabetic control group.

**Figure 9 biomedicines-11-01064-f009:**
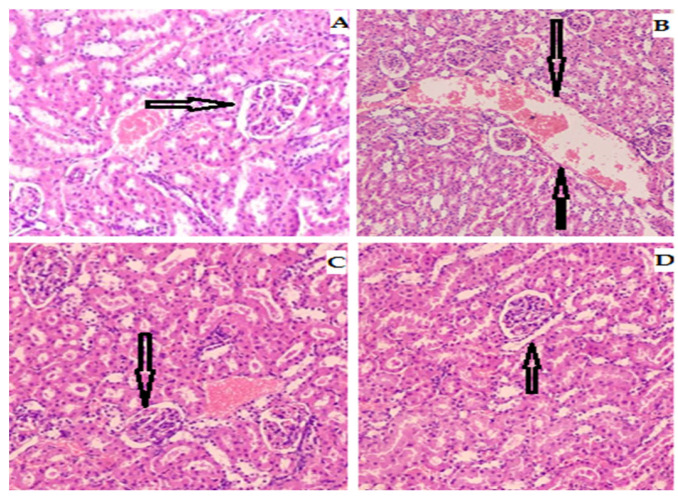
Photomicrographs of microsection of kidney of rat (**A**) control group showing the normal histological structure of renal parenchyma and bowman capsule, (**B**) diabetic group showing focal necrosis of epithelial lining renal tubules with architecture disturbance, (**C**,**D**) Rp-Nc 0.5 mg and 1 mg treated groups showing slight hydropic degeneration along restoring histological structure of hepatic lobules, Rp-Nc 1 mg treated group showing slight congestion along restoration of normal histological structure of histological structure of renal parenchyma (H & E scale bar 25 µm).

**Figure 10 biomedicines-11-01064-f010:**
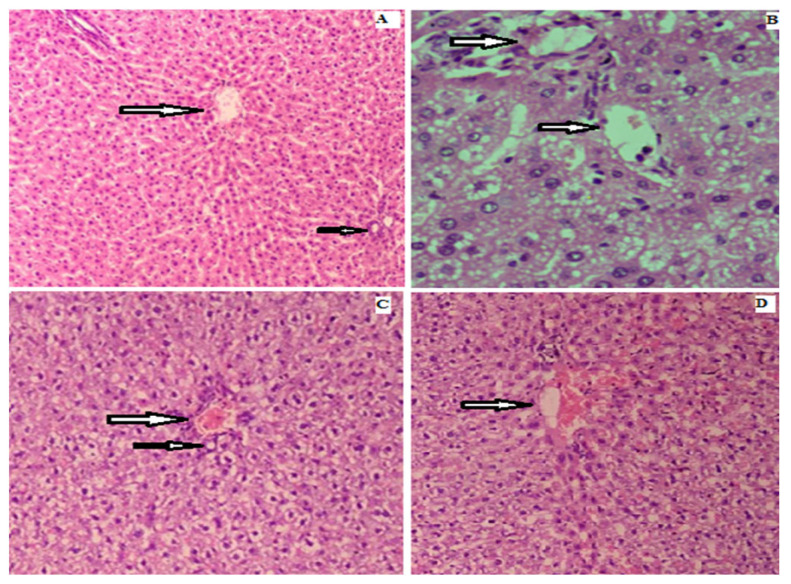
Photomicrographs of microsection of liver of rat (**A**) control group showing the normal histological structure of hepatic lobules and normal portal traid, (**B**) diabetic group showing focal necrosis and apoptosis of hepatocytes associated with mononuclear cell infiltration, (**C**) Rp-Nc 0.5 mg treated group showing slight hydropic degeneration along restoring histological structure of hepatic lobules, (**D**) Rp-Nc 1 mg treated group showing restoration of normal histological structure of hepatic lobules and normal portal traid (H & E scale bar 25 µm).

**Figure 11 biomedicines-11-01064-f011:**
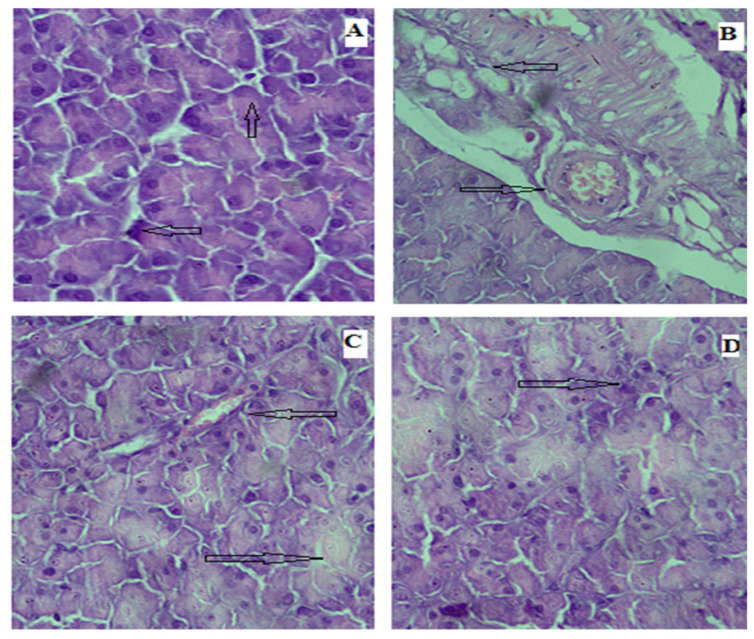
Photomicrographs of microsections of the pancreas of rats: (**A**) control group showing the normal histological structure of islets of Langerhans with normal pancreatic acini; (**B**) diabetic group showing ruptured and destroyed islets of Langerhans with damage in beta cells; and (**C**,**D**) Rp-Nc 0.5 mg- and 1 mg-treated groups showing some normal islets of Langerhans with some mild destruction in normal pancreatic ducts and in between normal pancreatic acini (H&E scale bar, 25 µm).

**Table 1 biomedicines-11-01064-t001:** Fabrication of Repaglinide nanosuspension.

Stabilizer/Surfactants	Concentration Stabilizer/Surfactants (%age *w*/*w*)	Flow Rate (µL/min) of Solvent/Antisolvent	Mean Particle Size (nm)/PDI	PDI
HPMC 15cps	1	50:80	960 ± 44	0.89 ± 36
PVP-K30	1	50:80	867 ± 21	0.94 ± 30
PVA	1	50:80	169 ± 34	0.32 ± 34
HPMC + PVP-K30	0.5 + 0.5	50:80	902 ± 44	1 ± 45
HPMC + P-F127	0.5 + 1	50:80	766 ± 54	1 ± 22
HPMC + PVA	0.5 + 0.5	50:80	937 ± 52	1 ± 43
PVA + PVP-K30	1 + 1	50:80	633 ± 35	1 ± 24
PVA + P-F127	1 + 2	50:80	413 ± 33	0.9 ± 33
PVA + P-F68	1 + 2	50:80	783 ± 31	0.682 ± 31

(Values are expressed as mean ± SEM). *w*/*w* = weight/weight.

**Table 2 biomedicines-11-01064-t002:** Optimization and Effect Flow rate on P.S/PDI of Repaglinide.

Concentration of PVA 1% (%ag *w*/*w*)	Flow Rate (µL/min) of Solvent/Antisolvent	Mean Particle Size (nm)/	PDI
1	50:50	245.32 ± 31	0.501 ± 22
-	50:60	223.44 ± 31	0.33 ± 32
-	50:70	209.91 ± 32	0.32 ± 43
-	50:80	180.33 ± 22	0.32 ± 34
-	50:90	134.31 ± 12	0.12 ± 13
-	50:100	113.99 ± 13	0.13 ± 23
-	50:110	91.43 ± 25	0.099 ± 22
-	50:120	71.31 ± 11	0.072 ± 12
-	50:130	77.25 ± 23	0.090 ± 22
-	50:140	81.33 ± 14	0.13 ± 34
-	50:150	141.13 ± 32	0.17 ± 41
-	50:160	199.43 ± 26	0.32 ± 32

(Values are expressed as mean ±SEM).

**Table 3 biomedicines-11-01064-t003:** Effect of Rp-Nc on blood glucose level.

Blood Glucose Level/Day (mg/dL)						
Groups/Treatment		1st	7th	14th	21st	28th
Normal control		104.34 ± 3.89	102.67 ± 3.66	105.80 ± 3.89	109.04 ± 3.61	107.22 ± 3.50
Diabetic control		498.07 ± 4.91 ^!!!^	501.45 ± 4.70 ^!!!^	504.39 ± 4.91 ^!!!^	490.72 ± 5.02 ^!!!^	486.65 ± 4.78 ^!!!^
Rp-Nc	0.5	503.09 ± 5.11	397.38 ± 4.86 **	276.90 ± 4.67 ***	170.81 ± 4.71 ***	124.05 ± 3.07 ***
	1	499.37 ± 4.89	371.41 ± 4.59 **	239.75 ± 4.82 ***	142.49 ± 4.66 ***	115.54 ± 3.02 ***
Rp (API)	1	507.06 ± 5.09	412.16 ± 4.93 *	294.61 ± 4.79 **	167.05 ± 4.58 ***	121.42 ± 3.44 ***
Tab (Mar)	1	497.46 ± 4.96	409.68 ± 5.09 *	298.12 ± 4.88 **	171.59 ± 4.71 ***	120.70 ± 3.60 ***

Mean ± SEM (*n* = 8). Significance was determined statistically using one way ANOVA following Dunnett’s comparison test, ^!!!^ *p* < 0.001 vs. normal control group and (* *p* < 0.05, ** *p* < 0.01, *** *p* < 0.001, *n* = 8) by vs. diabetic control group.

**Table 4 biomedicines-11-01064-t004:** Effect of Rp-Nc on body weight of animals.

Changes in Weight of Animals/Day						
Groups/Dose mg/kg		1st	7th	14th	21st	28th
Normal control		190.11 ± 4.78	187.91 ± 4.71	192.08 ± 4.91	191.96 ± 4.89	196.41 ± 5.11
Diabetic control		188.77 ± 5.10	182.69 ± 4.95 ^!!!^	176.21 ± 4.91 ^!!!^	158.93 ± 4.70 ^!!!^	147.16 ± 4.89 ^!!!^
Rp-Nc	0.5	187.62 ± 4.97	187.02 ± 4.77 *	184.21 ± 4.69 **	180.61 ± 4.91 ***	176.44 ± 4.90 ***
	1	190.30 ± 4.42	188.21 ± 4.89 *	186.05 ± 5.10 **	179.67 ± 4.82 *	182.91 ± 4.88 **
Rp (API)	1	186.59 ± 4.82	190.11 ± 4.94 *	179.88 ± 4.98 **	177.29 ± 4.79 *	173.86 ± 4.78 ***
Tab (Mar)	1	191.03 ± 4.51	189.71 ± 4.94 *	182.55 ± 4.79 **	183.08 ± 4.74 ***	179.50 ± 4.68 *

Mean ± SEM (*n* = 8). Significance was determined statistically using one way ANOVA following Dunnett’s comparison test, ^!!!^ *p* < 0.001 vs. normal control group and (* *p* < 0.05, ** *p* < 0.01, *** *p* < 0.001, *n* = 8) by vs. diabetic control group.

**Table 5 biomedicines-11-01064-t005:** Antihyperlipidemic effects of Rp-Nc.

Groups/Dose mg/kg		Total CH (mg/dL)	LDL (mg/dL)	TG (mg/dL)	HDL (mg/dL)
Normal control		68.25 ± 3.19	28.91 ± 1.91	71.98 ± 2.97	36.81 ± 2.48
Diabetic control		235.49 ± 4.03 ^!!!^	112.50 ± 2.73 ^!!!^	196.33 ± 4.10 ^!!!^	20.11 ± 1.73 ^!!!^
RpNc	0.5	78.09 ± 3.24 ***	32.04 ± 1.61 ***	84.17 ± 3.71 ***	29.01 ± 1.87 ***
	1	74.40 ± 3.11 ***	30.11 ± 1.87 ***	76.06 ± 2.96 ***	33.10 ± 2.29 ***
Rp (API)	1	81.11 ± 3.19 ***	29.93 ± 1.94 ***	85.37 ± 3.29 ***	28.03 ± 2.01 ***
Tab (Mar)	1	80.70 ± 3.06 ***	28.55 ± 2.01 ***	86.03 ± 3.11 ***	27.55 ± 1.88 ***

Mean ± SEM (*n* = 8). Significance was determined statistically using one way ANOVA following Dunnett’s comparison test, ^!!!^ *p* < 0.001 vs. normal control group and (*** *p* < 0.001, *n* = 8) by vs. diabetic control group.

**Table 6 biomedicines-11-01064-t006:** Effect of Rp-Nc on serum insulin and HbA1c level.

Groups/Dose mg/kg		Insulin Level (μU/mL/day)		HbA1c(%)
		0	28	--
Normal control		17.11 ± 0.29	18.05 ± 0.44	4.11
Diabetic control		8.91 ± 0.48	7.98 ± 0.37 ^!!!^	10.61 ^!!!^
Rp-Nc	0.5	8.74 ± 0.51	18.82 ± 0.69 ***	4.32 ***
	1	8.94 ± 0.68	18.63 ± 0.41 ***	4.21 ***
Rp (API)	1	8.78 ± 0.39	17.84 ± 0.51 ***	5.02 ***
Tab (Mar)	1	8.70 ± 0.45	18.01 ± 0.71 ***	4.87 ***

Mean ± SEM (*n* = 8). Significance was determined statistically using one way ANOVA following Dunnett’s comparison test, ^!!!^ *p* < 0.001 vs. normal control group and (*** *p* < 0.001, *n* = 8) by vs. diabetic control group.

**Table 7 biomedicines-11-01064-t007:** Effect of Rp-Nc on liver biomarkers.

Groups/Dose mg/kg		(ALP) IU	SGPT	Urea	Serum Creatinine (mg/dL)
Normal control		118.60 ± 2.86	60.48 ± 3.91	21.30 ± 1.88	0.63 ± 0.24
Diabetic control		238.54 ± 4.11 ^!!!^	97.62 ± 4.08 ^!!!^	316.54 ± 4.03 ^!!!^	3.22 ± 0.51 ^!!!^
Rp-Nc	0.5	152.82 ± 4.08 **	62.50 ± 3.70 ***	64.18 ± 2.78 ***	0.77 ± 0.48 ***
	1	147.17 ± 3.87 ***	59.47 ± 3.59 ***	61.30 ± 2.66 ***	0.74 ± 0.36 ***
Rp (API)	1	159.30 ± 3.72 **	64.58 ± 3.66 **	66.15 ± 2.71 **	0.80 ± 0.31 ***

Mean ± SEM (*n* = 8). Significance was determined statistically using one way ANOVA following Dunnett’s comparison test, ^!!!^ *p* < 0.001 vs. normal control group and (** *p* < 0.01, *** *p* < 0.001, *n* = 8) by vs. diabetic control group.

**Table 8 biomedicines-11-01064-t008:** Oxidative stress marker and antioxidant enzyme of various experimental groups.

Groups/Dose mg/kg		SOD (mU/mg Protein)	CAT (mU/mg Protein)	TBARS (nmol MDA/mg Protein)
Normal control		10.18 ± 0.61	22.07 ± 0.82	1.35 ± 0.28
Diabetic control		0.72 ± 0.11 ^!!!^	2.18 ± 0.29 ^!!!^	3.51 ± 0.43 ^!!!^
Rp-Nc	0.5	6.98 ± 0.48 ***	15.44 ± 1.03 ***	2.48 ± 0.38 **
	1	7.11 ± 0.81 ***	16.29 ± 1.24 ***	2.13 ± 0.29 ***
Rp (API)	1	6.31 ± 0.57 **	15.20 ± 1.01 **	2.55 ± 0.37 **
Tab (Mar)	1	6.69 ± 0.71 **	14.97 ± 1.08 **	2.51 ± 0.41 **

Mean ± SEM (n = 8). Significance was determined statistically using one way ANOVA following Dunnett’s comparison test, ^!!!^ *p*< 0.001 vs. normal control group and (** *p* < 0.01, *** *p* < 0.001, *n* = 8) by vs. diabetic control group.

## Data Availability

All the available data incorporated in the MS.

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
