# Peer review of "Formulation and Optimization of Repaglinide Nanoparticles Using Microfluidics for Enhanced Bioavailability and Management of Diabetes"

_biomedicines, 2023, doi:10.3390/biomedicines11041064_

Round 1

Reviewer 1 Report

This manuscript by Ahmed et al. demonstrates the preparation and optimization of Repaglinide, an anti-hyperglycemic drug with low bioavailability. Microfluidics has been explored for Rp preparation, which shows an improved release profile and superior particle characteristics. The authors evaluate Rp in vivo as well. Though the in vivo studies are supported by data, certain sections require clarity. This manuscript requires re-phrasing, proofreading, and grammar correction. 

Please see below for specific feedback:

The authors need to write the manuscript in scientific language. The manuscript also required proofreading prior to submission.

Line 3: Replace “enhance” with “enhanced”

Line 17: Replace with “poorly water-soluble ”medications with drugs.

Line 24: What is “treated Rp”?

Line 25: What is “processed Rp”? Authors must keep their language consistent.

Line 27: “considerably lower”? and please correct “P0.05”

Line 28: Replace “considerably” with “significantly”

Line 29: nanocyrstals at the “0.5”?

Line 30: Spelling of “dose”

The abstract needs to be edited for sentence formation and spellings. Authors should use scientific language.

Line 45: including as cardiovascular

Line 95: “was kindly gifted for research”

The authors need to elaborate on the preparation method for Repaglinide nanosuspension. How many nanosuspensions were prepared?

How did authors decide on the concentrations of stabilizers/ surfactants used?

What flow rates were used?

What was the authors’ protocol for particle size and zeta analyses?

What is “raw” Rp? Do the authors mean the free drug? Or clinically used drug?

Methods followed for DSC are missing

How was the formulation optimized?

How was the dissolution medium chosen?

What were the doses? Line 196 reference is missing

Line 216: Animals were “sacrificed”…. in a “humane” manner. Not “put to death”

The authors talk about stability studies, however, what is their acceptance criteria of stability?

Section 2.3: The sentence formation is quite confusing. Authors should re-phrase and have the section proofread.

Why is pH written as PH?

Section 2.5: What were the doses? These need to be mentioned in this section and not just referenced in the table or figures.

Section 2.7: How many animals were used for the study? Was n=8 distribution equal per group? How many groups were there? Was there any mortality?

Line 222: What is SEM?

Line 239: What is 80K? Please elaborate that this magnification. The spelling “below”

Table 1&2: Why is the flow rate represented as a ratio and not in uL/min?

Line 279: Change to “reported to be paramount for physical stability”

Line 283: What are the units? mV

Figure 2: What are the statistical differences or no differences? These need to be explained better.

Stability study for 40C shows that particle size and size distribution is increasing over time, hence particles are not stable at 40C temperature. Why have the authors mentioned that particles are stable?

Table 3: The authors have represented statistical significance by the same *** representation. And the results do not seem to be have high significant difference, however, this has not been explained well. Same for Fig. 7.

The authors need to edit this manuscript heavily before it is suitable for publication.

Reviewer 2 Report

The authors presented an interesting study about the Formulation and optimization of Repaglinide using microfluidics for enhances bioavailability for management of diabetes. 

The authors have concluded the findings of the present study indicated that nanocrystals of Rp with improved anti-diabetic properties and a comparable safety profile may be successfully created using microfluidic technology, a cutting-edge drug delivery system (DDS) approach.

Manuscript is well structured and study design and statistics are more appropriate.

However, there are some essential issues that need to be addressed so the manuscript would be of enough scientific quality for publication.

There are no adequate details about Repaglinide including its drawback/side effects towards vital organs and must update the recent citations.

Induction of diabetes: Authors should clarify whether, type 1 or type 2 diabetes?.

The author should define the abbreviations at their first use because some abbreviations were not defined.

Figure 2, need to improve the quality of figures including the size of the letters.

In Table 8, authors have studies about oxidative stress marker and antioxidant enzyme of various experimental groups but there are not complete results so I would suggest including the CAT and GPx etc.

Pancreatic images are not clear and staining protocol also not consistent so authors must repeat the staining for all pancreatic tissues and include the new images along with scale bar.

Round 2

Reviewer 1 Report

The authors have made significant improvements to their previous version of the manuscript, however, some obvious editing is still needed. Please see the corrections needed.

The authors are advised to once again review and polish their manuscript based on the suggestions below.

The abbreviations for Differential Scanning Calorimetry (DSC) and Power X-Ray Diffraction (P-XRD) can be stated at the beginning of the manuscript wherever mentioned first. After that, you can use just abbreviations for the rest of the manuscript. Same for other abbreviations.

Line 27 and 37: Change the double .. to a single .  (Punctuation)

Line 39: Why are there 3 p-values for 2 parameters?

Line 61: What is BCS. Given the elaboration of the abbreviation first and then the abbreviation can be used.

Page 76: Please correct “route’ spelling…..”reported to be one of…”

Line 79: Correct the spacing and double ..

Line 80…Change the double .. to singe .

Use correct spacing.

Line 82: spelling of “their”

Lines 86, 91, 97, 106, 107, 113: Same corrections as above. Please correct the spacing and use of ..

Line 270: Flow rates are usually represented as mL/min or uL/min. Use of 50 uL or 80 uL is representative of volume, not flow rate. If this is ul/min, then please mention as 50 uL/min not 50 uL.

Lines 422-435: What is the unit of blood glucose level? “found to be 276.90±4.67…..” Please mention units.

Tables 3 and 4: Authors still need to correct their P-values significance. The table mentions ** but there is no ** in the caption below in lines 441-443. For both P<0.01 and P<0.001, authors have represented ***. One of them needs to be **. Please correct the statistics.

Tables 5-8: Same as above. Does *** represent P<0.01 or P<0.001? There needs to be some distinction. Seems like the authors have copied and pasted the sentences from above.

Figures 7 & 8: Same as above. Please correct the stats.

Reviewer 2 Report

The authors have addressed most of the queries, but I would suggest authors to replace all the Photomicrographs of microsection of vital organs (kidney, pancreas and liver) since present pictures staining’s are not good and consistent.   

Figure 9. Photomicrographs of microsection of vital organs (kidney, pancreas and liver) stained 513 with hematoxylin and eosin stain (H&E) of different groups taken microscopically (x40).

Round 3

Reviewer 2 Report

the manuscript is acceptable for publication